# Baking Symmetry into GFlowNets

**Jiangyan Ma**[*]
Peking University

**Emmanuel Bengio**
Recursion Pharmaceuticals

**Yoshua Bengio**
Mila, Université de Montréal, CIFAR

**Dinghuai Zhang**
Mila, Université de Montréal

## Abstract

GFlowNets have exhibited promising performance in generating diverse candidates with high rewards. These networks generate objects incrementally and aim to learn a policy that assigns probability of sampling objects in proportion to rewards. However, the current training pipelines of GFlowNets do not consider the presence of isomorphic actions, which are actions resulting in symmetric or isomorphic states. This lack of symmetry increases the amount of samples required for training GFlowNets and can result in inefficient and potentially incorrect flow functions. As a consequence, the reward and diversity of the generated objects decrease. In this study, our objective is to integrate symmetries into GFlowNets by identifying equivalent actions during the generation process. Experimental results using synthetic data demonstrate the promising performance of our proposed approaches.

## 1 Introduction

Generative Flow Networks (GFlowNets, Bengio et al. [2021a]) have been proposed as a method to generate a wide range of high-quality candidates. By learning a stochastic policy $\pi$ for generating discrete objects $x$, GFlowNets iteratively add simple building blocks to partial objects, resulting in diverse and high-scoring candidates. These networks have shown promising performance in various tasks such as diverse molecule generation, active learning, biological sequence design, graph combinatorial optimization, and probabilistic learning [Bengio et al., 2021a, Jain et al., 2022, Zhang et al., 2023, 2022, Deleu et al., 2022].

However, prior works mostly neglect the internal symmetries within the generation process, leading to redundant data representations. Recent theoretical findings highlight the potential for improved sample complexity by incorporating data symmetry [Tahmasebi and Jegelka, 2023], and several approximation strategies for invariance have been proposed in the context of Graph Neural Networks (GNNs) [Murphy et al., 2019, 2018, Hu et al., 2021, Shuaibi et al., 2021, Puny et al., 2021, Ma et al., 2023]. Unfortunately, the current training pipelines for GFlowNets do not consider the existence of symmetric states and actions. This oversight could result in increased sample complexity and potentially incorrect flow probabilities, ultimately impacting the diversity and average reward of the generated objects.

To address this issue, we present two approaches in the GFlowNet training process that enforce invariance to isomorphic states and actions. When faced with isomorphic states, we suggest using canonization techniques to reduce the states (partial objects) to their canonical form, thereby reducing the size of the state space. For the graph generation process, isomorphic actions are defined as actions that lead to isomorphic preceding graphs, but these actions lack an efficient canonical form. In this scenario, we propose the use of handcrafted positional encodings (PE) to identify isomorphic

---

[*]Correspondence to: georgem@stu.pku.edu.cn

actions efficiently while maintaining expressive power. Our synthetic experiments demonstrate the effectiveness of these proposed approaches.

## 2 Background

We will examine the framework proposed by Bengio et al. [2021a], which involves a pointed directed acyclic graph (DAG) denoted as $(\mathcal{S}, \mathbb{A})$. In this setting, there is a designated initial state, $\mathcal{S}$ consists of a finite set of vertices called *states*, and $\mathbb{A} \subset \mathcal{S} \times \mathcal{S}$ represents a set of directed edges known as *actions*. If an action $\mathbf{s} \to \mathbf{s}'$ exists, we refer to $\mathbf{s}$ as the *parent* and $\mathbf{s}'$ as the *child*. Additionally, there is precisely one state that has no incoming edge, identified as the *initial state* $\mathbf{s}_0 \in \mathcal{S}$. States without outgoing edges are referred to as *terminating*. We denote the set of terminating states as $\mathcal{X}$. A *complete trajectory* is a sequence $\tau = (\mathbf{s}_0 \to \cdots \to \mathbf{s}_n)$ such that each $\mathbf{s}_i \to \mathbf{s}_{i+1}$ is an action and $\mathbf{s}_n \in \mathcal{X}$. We represent the set of complete trajectories as $\mathcal{T}$, and $\mathbf{x}_\tau$ indicates the last state of a complete trajectory $\tau$.

GFlowNets belong to a category of models that amortize the cost of sampling from an intractable target distribution over $\mathcal{X}$. These models accomplish this by learning a functional approximation of the target distribution using its unnormalized density or reward function denoted as $R: \mathcal{X} \to \mathbb{R}^+$. Bengio et al. [2021a] defines a trajectory flow $F: \mathcal{T} \to \mathbb{R}_{\geq 0}$. We can define a state flow $F(\mathbf{s}) = \sum_{\tau \ni \mathbf{s}} F(\tau)$ for any state $\mathbf{s}$ and an edge flow $F(\mathbf{s} \to \mathbf{s}') = \sum_{\tau \ni \mathbf{s} \to \mathbf{s}'} F(\tau)$ for any edge $\mathbf{s} \to \mathbf{s}'$. The trajectory flow induces a probability measure $P(\tau) = \frac{F(\tau)}{Z}$, where $Z = \sum_{\tau \in \mathcal{T}}$ represents the total flow. Furthermore, we define the forward policy $P_F(\mathbf{s}'|\mathbf{s}) = \frac{F(\mathbf{s} \to \mathbf{s}')}{F(\mathbf{s})}$ and the backward policy $P_B(\mathbf{s}|\mathbf{s}') = \frac{F(\mathbf{s} \to \mathbf{s}')}{F(\mathbf{s}')}$. In this context, the flows can be likened to the amount of water flowing through edges (similar to pipes) or states (resembling tees connecting pipes) [Malkin et al., 2022]. $R(\mathbf{x})$ represents the amount of water passing through the terminal state $\mathbf{x}$, while $P_F(\mathbf{s}'|\mathbf{s})$ corresponds to the relative quantity of water flowing in edges originating from $\mathbf{s}$.

### 2.1 GFlowNets training criteria

The objective of training GFlowNets is to enable the model to sample objects $x$ with a probability proportional to $R(\mathbf{x})$. To achieve this goal, we introduce several training criteria.

**Flow matching (FM).** We define a flow to be *consistent* if it satisfies the flow matching constraint for all internal states $\mathbf{s}$, meaning that the incoming flows equal the outgoing flows: $\sum_{\mathbf{s}'' \to \mathbf{s}} F(\mathbf{s}'' \to \mathbf{s}) = F(\mathbf{s}) = \sum_{\mathbf{s} \to \mathbf{s}'} F(\mathbf{s} \to \mathbf{s}')$. Bengio et al. [2021a] propose approximating the edge flow with a model $F_\theta(\mathbf{s}, \mathbf{s}')$ parameterized by $\theta$ using the FM objective. For non-terminal states, the objective is defined as $\mathcal{L}_{\mathrm{FM}}(\mathbf{s}) = (\log \sum_{(\mathbf{s}'' \to \mathbf{s}) \in \mathcal{A}} F_\theta(\mathbf{s}'', \mathbf{s}) - \log \sum_{(\mathbf{s} \to \mathbf{s}') \in \mathcal{A}} F_\theta(\mathbf{s}, \mathbf{s}'))^2$. At terminal states, a similar objective encourages the incoming flow to match the corresponding reward. The objective is optimized using trajectories sampled from a training policy $\pi$ with full support, such as a tempered version of $P_{F_\theta}$ or a mixture of $P_{F_\theta}$ with a uniform policy $U$: $\pi_\theta = (1 - \varepsilon)P_{F_\theta} + \varepsilon U$. This approach is similar to $\varepsilon$-greedy and entropy-regularized strategies in reinforcement learning (RL) to improve exploration. Bengio et al. [2021a] prove that if we reach a global minimum of the expected loss function and the training policy $\pi_\theta$ has full support, then GFlowNet samples from the target distribution.

**Detailed balance (DB).** The DB objective was proposed by Bengio et al. [2021b] to eliminate the need for computationally expensive summing operations over the parents or children of states. In DB-based learning, we train a neural network with a state flow model $F_\theta$, a forward policy model $P_{F_\theta}(\cdot|\mathbf{s})$, and a backward policy model $P_{B_\theta}(\cdot|\mathbf{s})$ parameterized by $\theta$. The optimization objective is to minimize $\mathcal{L}_{\mathrm{DB}}(\mathbf{s}, \mathbf{s}') = (\log(F_\theta(\mathbf{s})P_{F_\theta}(\mathbf{s}'|\mathbf{s})) - \log(F_\theta(\mathbf{s}')P_{B_\theta}(\mathbf{s}|\mathbf{s}')))^2$. Sampling from the target distribution is also done if a global minimum of the expected loss is achieved and $\pi_\theta$ has full support.

**Trajectory balance (TB).** The TB objective, proposed by Malkin et al. [2022], aims to enable faster credit assignment and learning over longer trajectories. The loss function for TB is $\mathcal{L}_{\mathrm{TB}}(\tau) = (\log(Z_\theta \prod_{t=0}^{n-1} P_{F_\theta}(\mathbf{s}_{t+1}|\mathbf{s}_t)) - \log(R(\mathbf{x}) \prod_{t=0}^{n-1} P_{B_\theta}(\mathbf{s}_t|\mathbf{s}_{t+1})))^2$, where $Z_\theta$ is a learnable parameter.

# 3 Method

We propose to enforce invariance constraints into the design of GFlowNets. Here, invariance of a function $f$ under a group of transformations $G$ is defined as $f(\mathbf{s}) = f(g \cdot \mathbf{s})$, where group element $g$ acts on the input domain of $f$. This means that inputs in the same orbit of $G$ are equivalent, in the sense that they produce the same output when fed to $f$.

## 3.1 State invariance

Now consider the case where the state space of GFlowNet $\mathcal{S}$ has a symmetric structure. Ideally we would require the forward policy of GFlowNet to satisfy *state invariance*:

$$P_F(\mathbf{s}'|\mathbf{s}) = P_F(g \cdot \mathbf{s}'|g \cdot \mathbf{s}), \quad \forall \mathbf{s}, \mathbf{s}' \in \mathcal{S}, \forall g \in G,$$

where $g$ is a transformation, such as a reflection in the $\mathcal{S}$ space. If this constraint could be achieved, then the GFlowNet would obtain improved performance since it will not waste capacity on unnecessary modeling. The same idea applies to flows, edge flows, and backward policies:

$$F(\mathbf{s} \to \mathbf{s}') = F(g \cdot \mathbf{s} \to g \cdot \mathbf{s}'), \; F(\mathbf{s}) = F(g \cdot \mathbf{s}),$$
$$P_B(\mathbf{s}|\mathbf{s}') = P_B(g \cdot \mathbf{s}|g \cdot \mathbf{s}'), \quad \forall \mathbf{s}, \mathbf{s}' \in \mathcal{S}, \forall g \in G.$$

**Symmetrization via group averaging**  When training a GFlowNet with flow matching, we parameterize the edge flow function $F(\mathbf{s} \to \mathbf{s}')$. One way to make the edge flow *output* correct, is to combine the output for equivalent actions, such as with averaging:

$$F(\mathbf{s} \to \mathbf{s}') = \frac{1}{|G|} \sum_{g \in G} \tilde{F}(g \cdot \mathbf{s} \to g \cdot \mathbf{s}').$$

We can use a similar strategy to deal with the forward and backward policy, which are needed in the detailed balance or trajectory balance parameterizations. For the state flow function $F(\mathbf{s})$, we can set $F(\mathbf{s}) = \frac{1}{|G|} \sum_{g \in G} \tilde{F}(g \cdot \mathbf{s})$ for an arbitrary neural network $\tilde{F}(\cdot)$. The same operation can be applied to the forward and backward policy modeling. The disadvantage of such procedure is a potentially high computational cost when the group $G$ is large, since enumerating all the group elements requires $\mathcal{O}(|G|)$ complexity.

**Symmetrization via canonical representation**  For some of the objects' representations, there is a more efficient way to detect symmetry. If there exists a function $\mathcal{C}(\cdot)$ (we call $\mathcal{C}(\mathbf{s})$ the *canonical form* of $\mathbf{s}$) such that

$$\mathcal{C}(\mathbf{s}) = \mathcal{C}(g \cdot \mathbf{s}), \quad \forall g \in G,$$

then we could achieve state invariance with an arbitrary neural network $\tilde{F}$ as follows:

$$F(\mathbf{s} \to \mathbf{s}') := \tilde{F}\big(\mathcal{C}(\mathbf{s}) \to \mathcal{C}(\mathbf{s}')\big).$$

The previous method needs $|G|$ forward passes of the neural network, while this method only requires one forward pass. The disadvantage of this procedure is that not all kinds of data possess a proper canonical form.

## 3.2 Action invariance

Symmetries may exist not only in the state space but also in the actions involved. Let's consider the graph generation environment, where actions include adding a node to an existing node and connecting them with a new edge. In Figure 1, we can see that there are three possible actions that lead from the left graph to the right graph. These actions involve adding a new node to any of the three existing nodes. We refer to them as *isomorphic actions* because they all result in the same (*i.e.*, isomorphic) graph. However, in the opposite direction, there is only one action that leads from the right graph to the left graph, which is the removal of node 4 and the edge connecting node 1 and node 4. Currently, GFlowNet training pipelines do not consider such isomorphic actions and would treat these three isomorphic actions as distinct, resulting in an incorrect forward edge flow that is three times larger than it should be.

In order to model the correct flow, it is necessary to identify symmetric actions and sum their probabilities during the training process. Since these symmetric actions do not have an efficient canonical form, it is essential to enumerate all of these actions given a graph-action pair.

**Symmetrization via isomorphism testing**
One straightforward method to enumerate isomorphic actions of a given action is to iterate through all possible actions and verify if they generate isomorphic graphs. However, performing direct isomorphism checks is computationally expensive and requires a time complexity

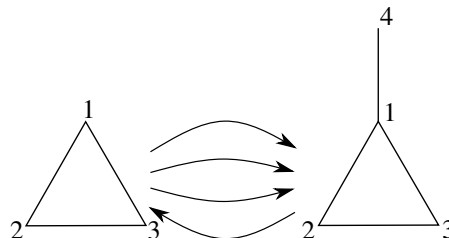

Figure 1: An example of action symmetry. There are three possible actions leading from the left graph to the right graph, but only one action leading backwards.

of $\mathcal{O}(n \times n!)$, as described by Cordella et al. [2001], which makes it impractical for larger graphs. Hence, in practice, we require approximation schemes to overcome this limitation.

**Symmetrization via graph-level positional encoding**  An alternative approach that improves efficiency over direct isomorphism testing involves using graph-level positional encoding (PE). PE functions as an embedding technique that maps a graph to a representation vector. Isomorphic graphs are guaranteed to have identical graph-level PEs. However, due to the inherent limitations in the expressive power of PE, it is possible for two non-isomorphic graphs to share the same graph-level PE. Our objective is to use graph-level PEs that are both expressive and computationally efficient, meaning they can effectively differentiate between most non-isomorphic graphs.

We investigate three kinds of PEs: PEs produced by the 1-WL test [Weisfeiler and Leman, 1968], random walk positional encoding (RWPE) [Li et al., 2020], and sum of edge features.

1. The 1-WL test is a color refinement method that finds for each node in each graph a signature based on the neighborhood around the node. These signatures can then be used to find the correspondance between nodes in the two graphs, which can be used to check for isomorphism.

2. The RWPE is defined as the concatenation of the diagonal elements of powers of the random walk matrix: $\boldsymbol{x}_i = [(\boldsymbol{A}\boldsymbol{D}^{-1})_{ii}^k], k = 1, 2, \ldots$, where $\boldsymbol{x}_i$ is the RWPE of node $i$ and $\boldsymbol{A}\boldsymbol{D}^{-1}$ is the random walk matrix of the graph. The original RWPE does not consider node colors, thus it cannot distinguish graphs with the same structure but different node colors. Here we propose to incorporate node colors into RWPE, by multiplying the powers of the random walk matrix with node colors: $\boldsymbol{x}_i = [(\boldsymbol{c}^{\mathrm{T}}\boldsymbol{A}\boldsymbol{D}^{-1})_i^k], k = 1, 2, \ldots$, where $\boldsymbol{c}$ is a vector representing node colors. We then take $\sum_i \boldsymbol{x}_i$ as the graph-level PE.

3. We could also calculate all edge features by summing[2] node features of their vertices and considering their sum as the graph-level feature. That is, for edge $(i, j)$, its edge feature is defined as $\boldsymbol{e}_{ij} \coloneqq \boldsymbol{x}_i + \boldsymbol{x}_j$ and we could take $\sum_{(i,j)\in\mathbb{E}} \boldsymbol{e}_{ij}$ as the graph-level PE.

We evaluate these three PE methods on graphs with a maximum of 7 nodes, where each node can be assigned one of two available colors. The actions considered in these graphs are the addition of nodes (*i.e.*, adding a node to an existing node and connecting them with an edge) and the addition of edges (*i.e.*, introducing a previously non-existent edge). We systematically test all possible combinations of graph-action pairs and assess the PE methods' ability to accurately enumerate all isomorphic actions. We record the error rate and running time for each PE method, and the results are summarized in Table 1.

As shown in Table 1, both "RW + edge" and "WL + RW + edge" achieves almost perfect accuracy in discovering isomorphic actions. However the former is much faster to compute than the latter, thus we use "RW + edge" as PE in our experiments.

**Symmetrization via node-level and edge-level positional encoding**  Symmetrization via graph-level PE requires iterating through all possible actions and calculating the graph-level PEs of each

---

[2]In practice we take the sum of squares instead of sum to avoid collision.

Table 1: The running time and error rate of different PE methods, where "WL" represents the 1-WL test feature, "RW" refers to random walk positional encodings (multiplied with node colors), and "edge" signifies the calculation of all edge features (by summing node features of their vertices) and considering their sum as the graph-level feature.

| PE method | Running time | Error rate |
|---|---|---|
| WL | 7h17min | 989650/2483411 = 0.3985 |
| WL + edge | 7h51min | 485123/2483411 = 0.1953 |
| RW | 4h24min | 1493301/2483411 = 0.6013 |
| RW + edge | 4h41min | 38/2483411 = 0.000013691 |
| WL + RW | 10h59min | 952227/2483411 = 0.3834 |
| WL + RW + edge | 10h56min | 0/2483411 = 0 |

preceding graph of these actions, which could be inefficient since we only care about actions on the single original graph. Here we propose an approximating alternative, by calculating the node-level and edge-level PEs of the original graph. If we wish to verify whether two node-adding actions are isomorphic, we can calculate the node-level PEs of their corresponding nodes and check whether they are identical. Similarly, we verify whether two edge-adding actions are isomorphic by checking whether the edge-level PEs of their corresponding edges are identical. In this way, we only need to calculate the PEs of nodes and edges of the original graph, which is more efficient than calculating the graph-level PEs of all preceding graphs. The node-level and edge-level PEs are the same as described above, without taking their sum as graph-level feature.

One disadvantage of using node-level and edge-level PEs is that actions with distinct PEs do not always lead to non-isomorphic graphs. We give a counterexample in Figure 2. As shown in Figure 2, edge (0,5) and edge (2,4) are structurally dissimilar, thus they have different PEs, yet they lead to isomorphic graphs. Luckily such counterexamples are quite rare (1 in a few thousand graphs) and they have negligible impact on model's performance. We use node-level and edge-level PEs in our experiments.

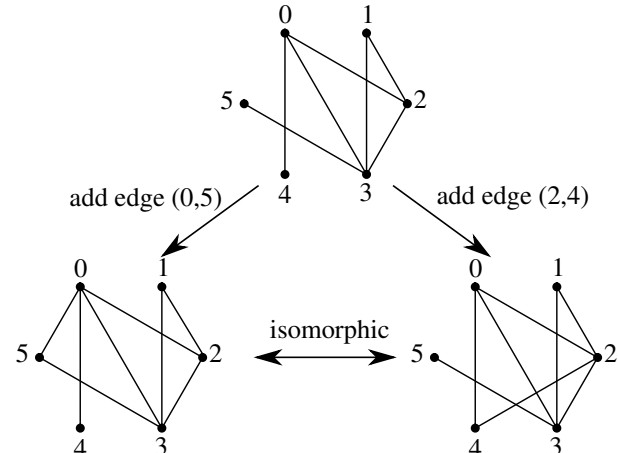

Figure 2: An example where two actions with different edge-level PEs could lead to isomorphic graphs.

## 4 Experiments

### 4.1 Experiments on state invariance

We implement and evaluate our method on the benchmark adopted by Bengio et al. [2021a], Malkin et al. [2022], namely hypergrid exploration sampling tasks. Starting from one corner, an agent moves in a grid-like world to explore the landscape defined by the following reward function in Equation equation 1. The agent starts at the fixed down left corner in every episode, and is only allowed to move up or right at each step. A third stop action indicates to terminate the trajectory and leave the agent at the grid it stands. Ideally, the agent should learn to stop at relatively high reward regions at terminating times. Also, we do not want the agent's solution to converge to a particular mode, but would like it to discover all different modes (4 modes in example in Figure 3). Therefore, we use the L1 distance between the empirical distribution (sampled by the GFlowNet

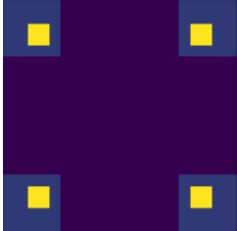

Figure 3: The HyperGrid environment.

agent) and the ground truth distribution as the evaluation metric.

$$R(\boldsymbol{x}) = R_0 + 0.5 \prod_{d=1}^{D} \mathrm{I}[0.25 < |x_d - 0.5|] + 2 \prod_{d=1}^{D} \mathrm{I}[0.3 < |x_d - 0.5| < 0.4]. \tag{1}$$

In Equation equation 1, $\boldsymbol{x} = (x_1, \dots, x_D)$ where $D$ is the number of state dimension. Notice that the reward function is invariant to the permutation of the coordinate ordering. This indicates that our flow model should be invariant to the permutation of state coordinates, *i.e.*, the group $G$ contains all permutations of $D$ elements. Notice that under such scenario, the proposed enumeration based method would require $D!$ times of neural network forward pass. On the other hand, in order to conduct the second canonization based method, we define

$$\mathcal{C}(\boldsymbol{x}) \coloneqq (x_{\pi(1)}, \dots, x_{\pi(D)}), \quad x_{\pi(1)} \leq \dots \leq x_{\pi(D)},$$

where $\pi$ denotes the permutation to arange the coordinates in descending order. Notice that on the states which contains two equal coordinates, our method cannot guarantee exact invariance and thus being an approximation (yet efficient) method.

We plot the L1 error with regards to the number of steps (*i.e.*, the number of visited states). The results in Figure 4 indicate that our proposed methods consistently outperform the original baseline, showing a better sample efficiency for symmetry involved methods.

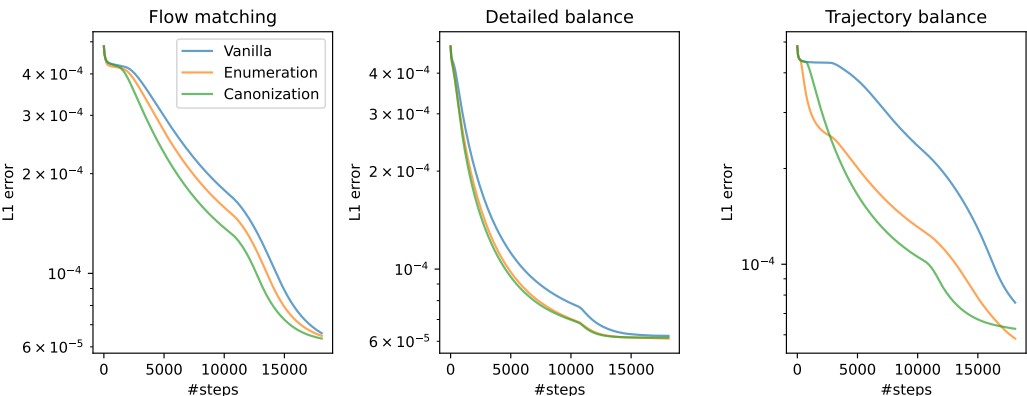

Figure 4: Results of the baseline, our method 1 (enumration based), and our method 2 (canonization based) on the hypergrid environment. We test with $\mathrm{horizon} = 16$, $\mathrm{dimension} = 3$, $R_0 = 0.001$.

## 4.2 Experiments on action invariance

To test the effects of action invariance, we setup a simple environment with all possible graphs of a maximum of 7 nodes, where the nodes can be one of two colors. We define three different reward functions with varying difficulty. The hardest function, **cliques**, requires the model to identify subgraphs in the state which are 4-cliques of at least 3 nodes of the same color. The second function, **neighbors**, requires the model to verify whether nodes have an even number of neighbors of the opposite color. Finally, the third function, **counting**, simply requires the model to count the number of nodes of each color in the state.

This environment has a total of 72296 states, which allows us to compute the ground truth probability $p(x)$ and the learned probability $p_\theta(x)$ relatively quickly. We compare three models, namely vanilla GFlowNet, GFlowNet with isomorphism testing (ground truth), and GFlowNet with positional encodings. The JS divergence between $p(x)$ and $p_\theta(x)$ are reported. We only report results on the counting reward and defer full experiment results to Appendix A.

As shown in Figure 5, incorporating action invariance improves the accuracy of the learned flow functions. Direct isomorphism testing and positional encodings achieve comparable performance, but the former has a time complexity of $\mathcal{O}(n \times n!)$ while the latter has a time complexity of $\mathcal{O}(n^3)$.

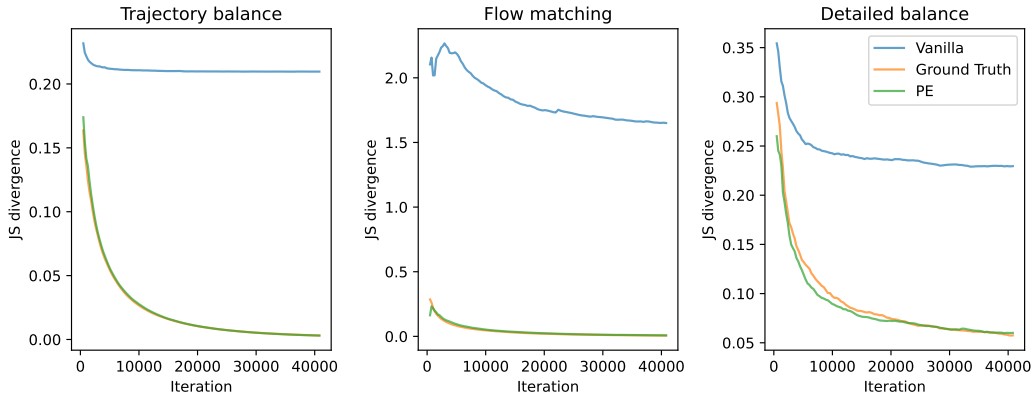

Figure 5: The JS divergence between $p(x)$ and $p_\theta(x)$ during offline GFlowNet training.

We also report the average reward during training in Figure 6. As shown in Figure 6, both direct isomorphism testing and positional encodings achieve better performance than vanilla GFlowNet, indicating more accurate flow probabilities indeed lead to higher rewards.

## 5 Conclusion

In this paper we propose to incorporate invariance to the internal symmetries within the generation process into GFlowNet training. For symmetric states, we propose enumeration-based and canonization-based symmetrization

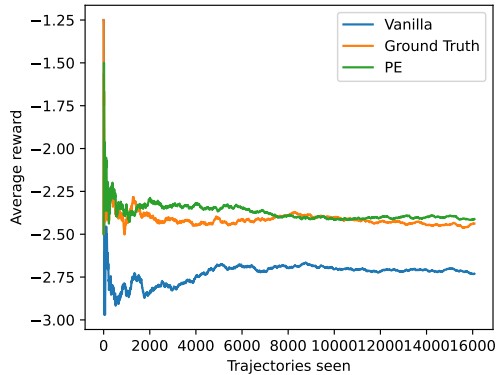

Figure 6: The average reward during training.

to achieve state invariance. For symmetric actions, we propose direct isomorphism testing and positional encodings as an efficient alternative. Results on synthetic experiments validate the efficacy of our proposed approaches.

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

# A   Full experiment results

Results on the **counting** reward function are reported in Figure 7.

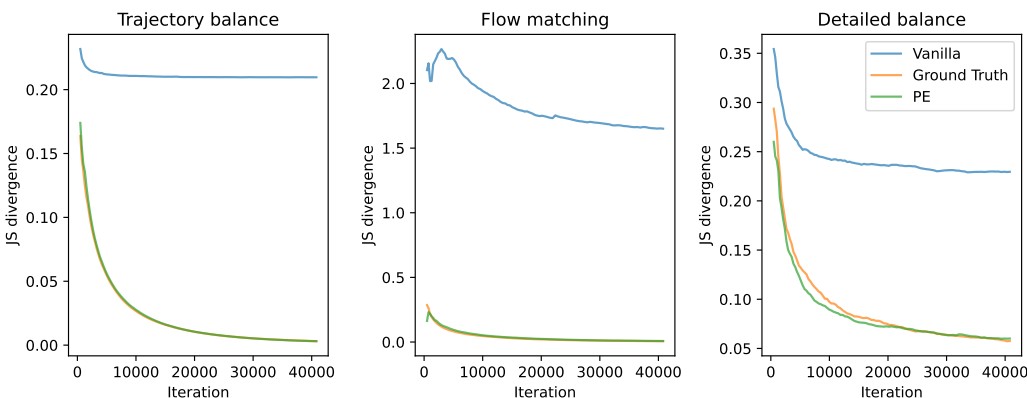

Figure 7: Results on the counting reward function.

Results on the **neighbors** reward function are reported in Figure 8.

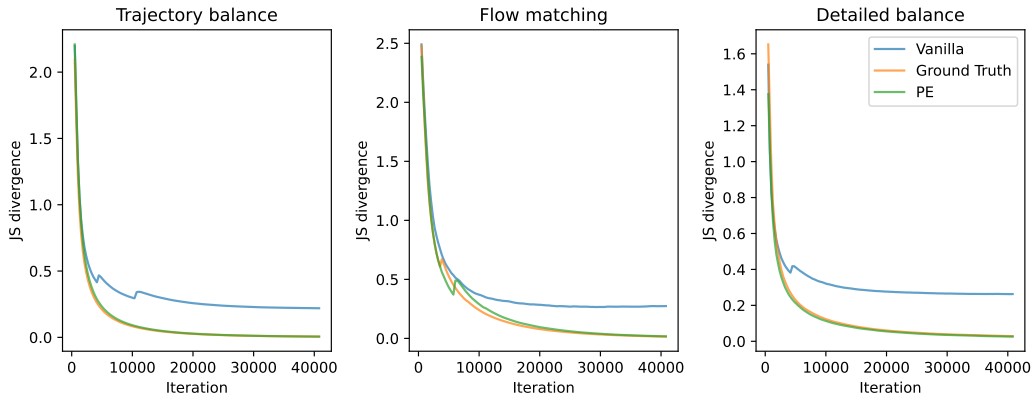

Figure 8: Results on the neighbors reward function.

Results on the **cliques** reward function are reported in Figure 9.

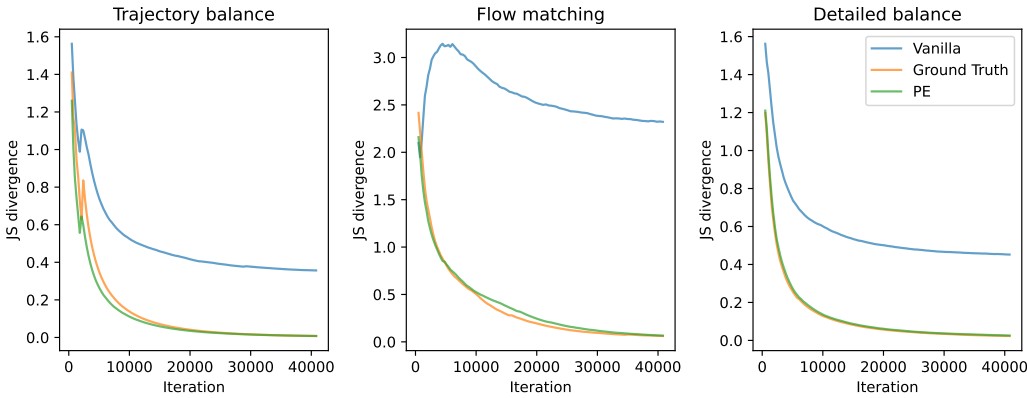

Figure 9: Results on the cliques reward function.

