# OpenReview forum: "Baking Symmetry into GFlowNets"
_NeurIPS.cc/2023/Workshop/AI4Science — NeurIPS2023-AI4Science Oral_

### Official Review · Reviewer_9w9b · 2023-10-15
**Enforcing Symmetry Invariance in GFlowNets**

**Rating:** 9
**Confidence:** 5

**Review:**

This paper proposes methods to incorporate symmetry invariance into Generative Flow Networks (GFlowNets) during training. For symmetric states, it suggests symmetrization via group averaging or canonical forms. For symmetric actions, it proposes direct isomorphism testing or efficient approximations with positional encodings. Experiments on synthetic data demonstrate the benefit of these techniques in improving sample efficiency and accuracy.

Quality - The work has clearly dived deeper into understanding the basic structure of GFlowNets and added ways to incorporate symmetrization into it.
Clarity - This is a clearly written paper describing a nice contribution to improve GFlowNets by handling symmetries
Originality - The proposed techniques for imposing state and action invariance are methodical and intuitive. The experiments provide a clear validation of the benefits on synthetic tasks, demonstrating improved sample efficiency and model accuracy. The methodology appears technically sound overall.
Significance - While the work is narrowly focused on symmetries in GFlowNets, it tackles an important issue that is likely relevant for other generative modeling scenarios as well. The enhancements seem generalizable, such as the one involving PEs for comparison of different actions and understanding symmetry.

---

### Official Review · Reviewer_RYXq · 2023-10-20
**The work has sufficiently high quality and novelty and is suitable to be published.**

**Rating:** 9
**Confidence:** 4

**Review:**

In this work, the authors propose to improve the training pipelines of GFlowNet by incorporating invariance to account for the internal symmetry which, if not considered, may result in inefficient and potentially incorrect flow functions. In particular, the authors use the canonical representation to detect the symmetry of isomorphic states and apply the graph-level, node-level, and edge-level positional encoding to efficiently detect the symmetry of isomorphic actions. The authors also kindly provide a systematic and comprehensive description of the principle underneath the symmetry incorporation scheme, which could reach out to more people inside or outside the area.

Considering the overall quality and novelty, it is the opinion of the reviewer that this work is highly suitable to be published in NeurIPS 2023 AI4Science Workshop, after some minor revisions.

1. The authors claim the improvement of incorporating action invariance on graphs with a maximum of 7 nodes. It is to the knowledge of the reviewer that graphs with such size have rather limited application in practical problems. A detailed explanation of such choice should be clarified.

2. It is reasonable to benchmark the improvement using the conceptual example in the original paper. However, it would be more convincing if the authors could use one more practical example to do the benchmark.